# Metabolomic Analysis of Carbohydrate and Amino Acid Changes Induced by Hypoxia in Naked Mole-Rat Brain and Liver

**DOI:** 10.3390/metabo12010056

**Published:** 2022-01-10

**Authors:** Hang Cheng, Yiming (Amy) Qin, Rashpal Dhillon, James Dowell, John M. Denu, Matthew E. Pamenter

**Affiliations:** 1Biology Department, University of Ottawa, Ottawa, ON K1N 6N5, Canada; hchen188@uottawa.ca; 2Department of Biomolecular Chemistry, University of Wisconsin-Madison, Madison, WI 53706, USA; qin26@wisc.edu (Y.Q.); john.denu@wisc.edu (J.M.D.); 3Wisconsin Institute for Discovery, University of Wisconsin-Madison, Madison, WI 53706, USA; dhillon@zoology.ubc.ca (R.D.); jadowell@wisc.edu (J.D.); 4Interdisciplinary Graduate Program in Nutritional Sciences, University of Wisconsin-Madison, Madison, WI 53706, USA; 5Morgridge Institute for Research, University of Wisconsin-Madison, Madison, WI 53706, USA; 6Brain and Mind Research Institute, University of Ottawa, Ottawa, ON K1H 8M5, Canada

**Keywords:** coenzyme, aspartic acid, glutamate, glutamine, glutathione, pentose phosphate pathway, dopamine, AMP, glycogen

## Abstract

Hypoxia poses a major physiological challenge for mammals and has significant impacts on cellular and systemic metabolism. As with many other small rodents, naked mole-rats (NMRs; *Heterocephalus glaber*), who are among the most hypoxia-tolerant mammals, respond to hypoxia by supressing energy demand (i.e., through a reduction in metabolic rate mediated by a variety of cell- and tissue-level strategies), and altering metabolic fuel use to rely primarily on carbohydrates. However, little is known regarding specific metabolite changes that underlie these responses. We hypothesized that NMR tissues utilize multiple strategies in responding to acute hypoxia, including the modulation of signalling pathways to reduce anabolism and reprogram carbohydrate metabolism. To address this question, we evaluated changes of 64 metabolites in NMR brain and liver following in vivo hypoxia exposure (7% O_2_, 4 h). We also examined changes in matched tissues from similarly treated hypoxia-intolerant mice. We report that, following exposure to in vivo hypoxia: (1) phenylalanine, tyrosine and tryptophan anabolism are supressed both in NMR brain and liver; (2) carbohydrate metabolism is reprogramed in NMR brain and liver, but in a divergent manner; (3) redox state is significantly altered in NMR brain; and (4) the AMP/ATP ratio is elevated in liver. Overall, our results suggest that hypoxia induces significant metabolic remodelling in NMR brain and liver via alterations of multiple metabolic pathways.

## 1. Introduction

The naked mole-rat (NMR, *Heterocephalus glaber*) is a eusocial rodent with many fascinating characteristics. Among these is an ability to tolerate acute or prolonged hypoxia, and even short-term anoxia [1,2,3]. In addition, a number of studies have demonstrated that various NMR tissues are tolerant to hypoxic and anoxic challenges ex vivo, with the research conducted to date focused primarily on the brain and heart [2,4,5,6]. While NMRs are thought to experience environmental hypoxia in nature, hypoxia is also a key component in a variety of pathophysiologies, including ischemic brain injury, heart failure, and non-alcoholic fatty liver disease, among others [7,8,9]. Thus, the ability of these mammals to tolerate systemic and tissue-level hypoxia without apparent detriment is of interest. 

Hypoxia poses a metabolic challenge at the cellular level: reduced O_2_ availability limits the rate of ATP production from mitochondrial oxidative phosphorylation (OXPHOS) [10]. If ATP demand is not similarly reduced to match ATP supply in hypoxia, cell death typically ensues [11]. It is generally accepted that metabolic pathways are altered in response to hypoxic stress; hence, the abundance of metabolic intermediates/products are likely to also be altered. These changes can be informative of cell pathway modulations in hypoxia because numerous metabolic intermediates/products have specific roles in modulating both beneficial and pathophysiological pathways. For example, anaerobic glycolysis is often upregulated to maintain ATP production when glucose oxidation is suppressed by limited O_2_ during tissue hypoxia. As a result, lactate, which is the primary end-product of anaerobic glycolysis and can be directly metabolized by certain cell types, is significantly elevated under these conditions. Other examples include alterations in carbohydrate metabolism (e.g., glucose, glycogen, and insulin-regulated pathways), which are inconsistently impacted by hypoxia in both hypoxia-tolerant and hypoxia-intolerant species [12,13,14,15], and which have major impacts on systemic function. 

Hypoxia can also lead to imbalances in important neurotransmitters and cellular second messengers. For example, excess extracellular glutamate accumulates during hypoxia in the brain of most hypoxia-intolerant mammals, which mediates excitotoxicity and can lead to deleterious reactive oxygen species (ROS) generation and cytotoxic increases in intracellular Ca^2+^ [10]. Similarly, succinate is involved in glutamate metabolism and the accumulation of succinate during periods of low O_2_ stress contributes to damage in heart, brain, and liver tissue both in hypoxia and following reoxygenation in hypoxia-intolerant species [16,17]. 

Although our understanding of metabolic responses to reduced O_2_ availability and related changes in metabolites is generally patchy in species adapted to hypoxia, there are several common patterns of hypoxia-induced metabolic remodelling that have emerged in champions of hypoxia-tolerance, such as goldfish, certain species of freshwater turtles, ground squirrels, and various African mole-rat species [1,18,19,20,21]. For example, at the whole-animal level, marked metabolic rate suppression is observed in many of these species during periods of acute or prolonged hypoxia or anoxia, which is achieved by a wide variety of organ-specific reductions in cellular and systemic function, such as decreases in excitatory neurotransmitter release, suppression of action potentials, and reduced ion permeability in brain as well as liver, etc. [11,19]. 

A similar response was documented in NMRs, who supress their whole animal metabolic rate by up to 85% in acute and severe hypoxia [5]. Systemically, blood glucose increases during hypoxia and NMRs undergo a whole-animal shift to carbohydrate fuel use [22]. Furthermore, NMR brain and heart can metabolize fructose ex vivo [2], and brain Na^+^/K^+^-ATPase activity is regionally regulated during in vivo hypoxia [23]. However, despite these advances, little is known regarding hypoxia-mediated changes in key signalling metabolites and metabolic intermediates in NMRs.

To address this knowledge gap, we aimed to comprehensively and systematically evaluate hypoxic changes in key signalling intermediates and metabolites to better understand the strategies employed by NMRs at the cellular level in response to hypoxia. Hence, we identified and analyzed hypoxia-induced differences in the metabolomic profiles of NMR and mouse brain and liver to reveal potential mechanisms that could explain the relatively superior tolerance of NMRs to hypoxia in these tissues. 

## 2. Result

To analyze differences in the identified metabolites (64 for each tissue, Appendix A) among species, organs, and/or treatments, we employed principle components analysis (PCA) and hierarchical clustering heatmap analyses using MetaboAnalyst 5.0 [24]. Hypoxia-induced metabolomic profile changes were well resolved in each tissue (Figure 1), and the hypoxia treatment accounted for 58.9% to 75.6% of sample variability (principal components 1 and 2). Furthermore, a hierarchical clustering heat map indicated good sample clustering by treatment within NMR and mouse tissues (Figure 2). Few metabolites were similarly changed between species and/or tissues; however, changes in metabolites were more numerable and of greater magnitude in NMRs than in mice (e.g., the number of metabolites that were significantly different between treatments was 24/64 in NMR brain and 29/64 in NMR liver, vs. 15/64 in mouse brain and 25/64 in mouse liver) (Figure 3). 

Few pathways were commonly altered during acute hypoxia within the same species or tissues; however, we observed differences between tissues and species in several key pathways. Generally, the reprogramming of metabolic pathways differed between NMR brain and liver relative to mouse following acute hypoxia. Overall, these changes suggest that: (1) aspartate and glutamate metabolism are enhanced in both NMR brain and liver; (2) pentose phosphate pathways are partially altered in NMR brain; (3) glutathione-related redox states are adjusted in NMR brain; and (4) glucose homeostasis is remodelled in NMR liver.

### 2.1. Hypoxia-Mediated Metabolomic Profile Changes Are Greater in NMR Than Mouse Brain

As shown in Figure 3A,B, there were similar trends and overlapping changes of metabolites between normoxic and hypoxic NMR and mouse brains. However, we identified more metabolites that were both down- and upregulated in hypoxic NMR brain. We assigned all significant metabolites to KEGG pathways using MetaboAnalyst 5.0 [24]. A total of 22 KEGG pathways were enriched in mouse brain, while 35 were enriched in NMR, including 19 of the 22 elevated in mice (Figure 4A,B, Appendix A). Furthermore, within the same pathways, a greater number of metabolites were identified as altered in NMR brain than mouse (e.g., metabolites related to aminoacyl-tRNA biosynthesis, valine leucine isoleucine biosynthesis, and arginine biosynthesis, etc.) (Figure 4A,B; note that ‘biosynthesis’ in this context is a general term for increased bioavailability of a given metabolite and may not be due to the actual synthesis of a given metabolite within the organism). Moreover, several pathways were altered only in NMRs with particularly low *p*-values and/or high impact factors, including nicotinate nicotinamide metabolism, glutathione metabolism, and d-glutamine/d-glutamate metabolism (Figure 4A,B). Overall, the major components of these pathways can be divided into three groups, which are amino acids, carbohydrates, and coenzymes.

Glutamate, glutamine, aspartate, arginine, ornithine, proline, threonine, isoleucine, leucine, and valine were identified as changing in both NMR and mouse brain; however, the number of significant changes was larger in hypoxic NMR brain (Figure 3A,B). Therefore, these metabolites contributed higher confidence (larger hit numbers and lower *p*-values) in arginine-related metabolism, valine, leucine, and isoleucine metabolism, and alanine, aspartate, and glutamate metabolism in NMR than mouse brain (Figure 4A,B). Phenylalanine and tyrosine were increased in hypoxic NMR brain, but not in hypoxic mouse brain (Figure 3A,B), and were assigned to phenylalanine metabolism (Figure 4A,B).

There were also changes in a group of metabolites that were not assigned using MetaboAnalyst, including a reduction in fructose 6-phosphate and accumulations of sedoheptulose 7-phosphate and glucose 6-phosphate in hypoxic NMR brain, which participate in the pentose phosphate pathway (Figure 3A,B).

The most significant changes in hypoxic NMR brain were the glutathione metabolism and nicotinate nicotinamide metabolism pathways (Figure 4A, Appendix A), as indicated by decreased levels of ornithine, glutamate, glutathione, aspartic acid, NAD^+^, and NADP^+^, and increased NADPH (Figure 3A). In contrast, these pathways were unchanged in mouse brain.

### 2.2. Metabolomic Profile Changes following Hypoxia Are Greater in Liver Than Brain in NMRs and Mice

More KEGG pathways were enriched by hypoxia in liver than brain in both NMRs and mice (Figure 4C,D, Appendix A). Specifically, 43 pathways were modified by hypoxia in mouse liver and 44 in NMR liver, and the NMR list contained 38 of the 43 altered pathways in mice. Similar to the changes in brain, several of the overlapping KEGG pathways in liver also included a greater number of metabolites, lower *p*-values, and/or higher impact factors in NMR than mouse (Figure 4C,D, Appendix A). Among these pathways, amino acids, carbohydrates, and purines were the major contributors.

As in NMR brain, an accumulation of phenylalanine and tyrosine occurred in hypoxic NMR liver but only phenylalanine accumulated in hypoxic mouse liver (Figure 3C,D). Moreover, different numbers and groups of amino acids were identified as changing due to hypoxia between NMR and mouse liver, but these were nonetheless assigned to the same pathway (Figure 3C,D and Figure 4C,D, Appendix A), including alpha-ketoglutarate, glutamine, aspartate, and arginine in NMR liver, and glutamine, fumarate, and ornithine in mouse liver, which were all assigned to arginine biosynthesis. Other pathways identified include alanine, aspartate, and glutamate metabolism, purine metabolism, phenylalanine metabolism, d-glutamine/d-glutamate metabolism, and valine, leucine, and isoleucine metabolism (Figure 4C,D, Appendix A).

Several metabolites decreased with hypoxia in NMR liver but not mouse liver. These were primarily related to carbohydrate metabolism, including citrate, alpha-ketoglutarate, fructose 6-phosphate, and glucose 1-phosphate (Figure 3C). Glycolysis/gluconeogenesis were the major pathways that were assigned to these metabolites (Figure 4C, Appendix A). Lactate increased in hypoxic NMR brain but not liver and decreased in hypoxic mouse liver (Figure 3A,C,D). Conversely, a hypoxic reduction in malate and fumarate was identified in mouse liver but not NMR liver, and this change was assigned to the pyruvate metabolism pathway (Figure 3D and Figure 4D).

Finally, purine metabolism and pyrimidine metabolism were also identified in both hypoxic NMR and mouse liver (Figure 4C,D); however, different purines were altered between NMR and mouse liver. For example, AMP increased and ATP decreased in NMR liver, whereas AMP was unchanged and ATP increased in mouse liver (Figure 3C,D).

## 3. Discussion

In this study, we identified and analyzed changes in metabolites following acute in vivo hypoxic stress in NMR and mouse brain and liver. Although some metabolites and metabolic pathways are similarly altered by hypoxia between species and tissues, our results highlight several major changes induced by hypoxia in NMR brain and liver but not in mouse. Generally, these changes are consistent with (1) the inhibition of energy demand and metabolic suppression in NMR brain, (2) a potential reduction in excessive glucose release to the peripheral circulation during hypoxia in NMR liver, and (3) the putative activation of AMP-activated protein kinase (AMPK) to regulate glucose homeostasis in NMR liver.

### 3.1. Metabolic Remodelling and Redox State Switching in NMR Brain during Acute Hypoxia

We identified a handful of metabolites that change similarly between NMR and mouse brain following in vivo hypoxia, including valine, leucine, isoleucine, glucose 6-phosphate, glutamine, and arginine. Beyond these common changes, NMR brain exhibits broad metabolic reprograming during hypoxia related to (1) glutamine and glutamate metabolism, (2) phenylalanine/tyrosine-related neurotransmitter synthesis, (3) pentose phosphate pathways, and (4) coenzyme-related metabolism.

Valine, leucine, and isoleucine accumulate with hypoxia in NMR and mouse brain and mouse liver. The accumulation of essential amino acids is a common response to stress in mammals, including to hypoxia [25]. Interestingly, an accumulation of threonine during hypoxia is observed in NMR brain and liver but not in mouse brain and liver. The mechanism underlying this change of threonine in NMR brain is unclear but may be related to AMPK activity in NMR liver (see below).

Aspartate and glutamate metabolism are critical for maintaining energetic balance in homeostatic situations and also drive excitotoxicity in the central nervous system. We observe a decrease in glutamate and glutamine in hypoxic NMR brain but only glutamine decreases in hypoxic mouse brain. Decreases in these excitatory amino acids may be neuroprotective in NMRs, as derangements in glutamate are associated with excitotoxic cell death in hypoxia-intolerant mammalian brain [26]. In support of this, previous studies have reported that NMR brain is largely tolerant to hypoxic and ischemic stress ex vivo [2,27,28], although the underlying protective mechanisms are largely unknown. It is possible that changes in the function of the aspartate–glutamate carrier, glutamic-oxaloacetic transaminase, and/or glutamate dehydrogenase occur during hypoxia in NMR brain, conferring some degree of neuroprotection. Alternatively, the reduction in lysine in NMR brain suggests that this tissue might have increased aspartate/glutamate/glutamine demand and/or metabolism, since lysine is an important precursor in the de novo synthesis of glutamate [29]. Further studies are warranted to explore the regulation of excitatory neurotransmission in NMR brain during hypoxia.

Phenylalanine and tyrosine are precursors to dopamine, which is, in turn, a precursor of epinephrine and norepinephrine [30]. We observe increases in both metabolites in the hypoxic NMR brain; however, the mechanism underlying these changes in phenylalanine and tyrosine is unclear. We speculate that NMR brain may reduce the synthesis of epinephrine and norepinephrine, leading to decreases in physical activity during hypoxia, which is likely most severe in crowded nest chambers [1]. Additionally, limited molecular O_2_ availability during hypoxia may affect the activity of dioxygenases that contribute to the metabolism of these amino acids [31].

Downregulation of fructose 6-phosphate is observed in hypoxic NMR brain, while interestingly, glucose 6-phosphate and sedoheptulose 7-phosphate are upregulated. It is thus likely that the pentose phosphate pathway is remodelled during hypoxia in NMR brain. This finding is similar to a previous report in anoxia [2].

Accompanying these changes of amino acids and carbohydrates, the redox state in NMR brain changes dramatically during hypoxia. This is apparent from the very large increase in NADPH and from reductions in FAD, NAD^+^, and NADP^+^. Increased NADH levels are essential to provide electrons for mitochondrial OXPHOS [32]. In this study, we report changes of nicotinate and nicotinamide metabolism in hypoxic NMR brain, which indicates that NMR brain mitochondria decrease OXPHOS function during hypoxia, as previously reported [5]. The source of increased NADPH is unknown; however, nicotinamide nucleotide transhydrogenase is a physiologically relevant source of mitochondrial NADPH that is activated when the mitochondrial electron transport chain is impaired, thus slowing the conversion of NADH to NAD [33]. This may occur during hypoxia when the efficiency of the electron transport chain becomes limited. There are also additional enzymatic oxidative reactions that generate NADPH and may be activated here (e.g., glutamate dehydrogenase). During normoxic conditions, most of the cytosolic NADPH is contributed by the oxidative pentose phosphate pathway, and most NADPH is used for both reductive biosynthesis and antioxidant defence [34]. Thus, the elevation of NADPH in NMR brain might also reduce NADPH-related antioxidant defence during hypoxia as a biproduct of the coordinated lowering of O_2_ consumption and ROS production in the brain and potentially across the whole organism, since oxidative metabolism can directly contribute to the generation of ROS [35]. Indeed, we recently reported that H_2_O_2_ consumption rates are around two- and five-fold greater in NMR than mouse, in skeletal muscle and heart mitochondria, respectively [36]. Thus, decreased ROS generation in hypoxia may reduce the activity of mitochondrial thioredoxin and glutathione scavenging pathways and, thus, contribute to the accumulation of NADPH.

### 3.2. Inhibition of Anabolism and Remodelling of Glucose Homeostasis in NMR Liver during Acute Hypoxia

There are two main hypoxic changes in NMR liver metabolic function that diverge significantly from changes in mouse liver: (1) amino acid-related biosynthesis (anabolic pathways) is altered, and (2) glucose homeostasis is remodelled.

First, and as in NMR brain, phenylalanine, tyrosine, and tryptophan-involved anabolic pathways are inhibited by hypoxia in NMR liver but not in mouse liver, which might indicate the suppression of ATP consuming metabolism in NMRs during hypoxia. Conversely, and as discussed above, glutamine decreases in both NMR and mouse liver with hypoxia, while aspartate and alanine increase in NMR liver but are not changed or decrease in mouse liver. Given the reductions in glutamine, glutathione, and alpha-ketoglutarate in NMR liver, we speculate that alanine might participate in regulating glutamate during hypoxia, as alanine aminotransferase can simultaneously catabolize and synthesize glutamate [37]. Furthermore, in the arginine biosynthesis pathway, alpha-ketoglutarate, glutamine, and arginine are decrease in NMR liver, while aspartate increases. In hypoxic mouse liver, glutamine and fumarate decrease, and ornithine increases. Our data suggest that arginine metabolism and glutamate metabolism interact with each other, but this requires further research.

We observe that fructose 6-phosphate, glucose 1-phosphate, and glucose 6-phosphate are all reduced but lactate is unchanged in NMR liver during hypoxia. These changes could relate to enhanced glycolysis and glycogen synthase activity, as we previously reported that liver glucose levels are significantly decreased following acute in vivo hypoxia [22]. However, we and others also reported that blood glucose increases in hypoxia and anoxia [2,22], changes that were previously attributed to the release of glucose from the liver, but which could also be due to decreased uptake of blood glucose to metabolically arrested tissues.

Although NMRs are reported to be glucose tolerant [38], the accumulation of alanine, a gluconeogenic amino acid, in the liver during hypoxia may reveal a possible mechanism as to how NMRs supress the release of glucose to the systemic circulation from the liver during hypoxia, which is by inhibiting the generation of glucose from precursors such as alanine. Another group of amino acids that are important for glucose regulation are the branched-chain amino acids valine, leucine, and isoleucine [39]. Valine, leucine, and isoleucine all increase in hypoxic mouse liver, but only leucine accumulates in hypoxic NMR liver. The mechanism of how leucine and isoleucine alter blood glucose is still under investigation, but some studies report that exogenous leucine and isoleucine can reduce blood glucose in either normal or diabetic rodents and via both insulinotropic and non-insulinotropic pathways [40,41,42]. Interestingly, NMRs are sensitive to acute insulin with a low dose and a long period [38]. Clearly, further work is warranted to investigate glucose metabolism in NMRs during hypoxia.

Other, perhaps critical, metabolic changes in NMR liver are related to purine and pyrimidine metabolism. Several such metabolites are reduced in hypoxic NMR liver, including CTP, ATP, GTP, and cytidine. Conversely CTP, GTP, UTP, ATP, ADP, and GDP accumulate in hypoxic mouse liver. Most intriguing is the accumulation of AMP and the reduction in ATP in hypoxic NMR liver, changes that are not observed in NMR brain or in mouse liver with hypoxic treatment. AMPK is a phylogenetically conserved serine/threonine protein kinase, which was proposed to play an important role in protecting cellular function under energy-restricted conditions [43], such as hypoxia. In hypoxia-intolerant species, insufficient O_2_ leads to energy deficits and, thus, a change in the cellular AMP/ATP ratio [44,45]. Consequently, AMPK is activated by this low energy status and acts to restore energy balance through the inhibition of ATP-consuming processes and the activation of ATP-generating processes [46]. AMPK activation in liver has been reported following chronic (>48 h) hypoxia or starvation [47,48] but rarely following acute short-term hypoxia (a few hrs) in mice or rats. Interestingly, the inhibition of AMPK activation under acute hypoxia was recently reported in NMR muscle [49], but indirect evidence suggests that it may be upregulated in the brain [50], suggesting region-specific regulation of this enzyme in the hypoxic NMR.

Thus, the changes of AMP and ATP in NMR liver might elevate AMPK activity during acute hypoxia. This mechanism could drive the increased level of methionine and threonine, which indicates that methionine/threonine catabolism or related protein synthesis is reduced, because threonine-related anabolic pathways are inhibited by AMPK activity [51]. However, there are other aspects of this system that require further study. For example, pharmacological activation of AMPK inhibits glycogen synthesis in rat liver [52]; however, mouse liver lacking AMPK also has lower levels of glycogen [53]. On the other hand, liver-specific AMPK activation may offer an effective therapeutic strategy for protecting against diet-induced obesity by increasing fat oxidation in rodent models [54,55]. Notably, NMR liver is reported to have a higher reliance on/capacity to utilize fatty acids for energy production through a greater abundance of lipid turnover-enzymes [56]. Taken together, our data suggest that the AMPK pathway in NMR liver is a potential target for future investigation, including whether and how its activity is altered during acute hypoxia and what subsequent metabolic processes are modulated by putative AMPK activation during hypoxia in this tissue.

## 4. Methods

### 4.1. Animals and In Vivo Treatments

Subordinate NMRs (1–2 years old) were group-housed in interconnected multi-cage systems at 30 °C and 21% O_2_ in 50% humidity with a 12L:12D light cycle. NMRs were fed fresh tubers, vegetables, fruit, and Pronutro cereal supplement *ad libitum*. Male CD-1 mice (12–14 weeks old) were obtained from Charles River and housed at room temperature under a 12L:12D light cycle and fed rodent chow *ad libitum*. Animals were not fasted prior to experimental trials and were thus in a fed state. Animals were not provided with food during the treatment periods. 

NMRs that are housed within colony systems do not exhibit the circadian rhythmicity of general locomotor activity [57,58]. Therefore, our results are not likely to be confounded by circadian rhythms in this species. Nonetheless, all animals were treated and sacrificed at the same time of day to limit any impact of this rhythmicity on our results. Furthermore, non-breeding (subordinate) NMRs do not undergo sexual development or express sexual hormones and, thus, we did not take sex into consideration when evaluating our results [59]. Both NMRs (30 °C) and mice (24 °C) were exposed to one of two treatment conditions: (1) normoxia (21% O_2_, balance N_2_; *n* = 4), or (2) acute hypoxia (4 h in 7% O_2_, balance N_2_; *n* = 4). Immediately following treatment, animals were sacrificed by cervical dislocation followed by immediate decapitation. Brains and livers were rapidly dissected within 30 s of euthanasia, and immediately flash frozen in liquid nitrogen and then stored at −80 °C until analysis. We chose 7% O_2_ as the hypoxic treatment to ensure a robust hypoxic challenge to both experimental species and because most previously published molecular and physiological data examining mechanisms of adaptation to hypoxia in NMRs are from similarly treated animals.

### 4.2. Metabolite Extraction and LC-MS-Based Metabolite Profiling

The metabolite extraction method was adopted from [60] with modifications. Briefly, tissues were pulverized prior to metabolite extraction. Pulverized tissue (~30 mg) was incubated with 1 mL of ice-cold 8:2 methanol:water solution for 5 min on dry ice after 15-s of vortexing. Tissue homogenates were centrifuged at maximum speed (21,100× *g*) for 5 min at 4 °C and the supernatant was collected into a new tube. The remaining pellet was loosened using a pipette tip and incubated with 400 µL of ice-cold 4:2:2 methanol:acetonitrile:water solution for 5 min on ice. The tissue homogenate was centrifuged again at maximum speed for 5 min at 4 °C and the supernatant was pooled with the first metabolite extraction. The 4:4:2 methanol:acetonitrile:water extraction was then repeated, and three extractions were pooled and completely dried using a SpeedVac (Thermo Fisher Savant ISS110) with nitrogen flow at room temperature. The dried metabolite samples were resuspended in water (150 µL water per 5 mg frozen tissue) and centrifuged at maximum speed for 5 min at 4 °C. The supernatant was used for LC-MS metabolite profiling.

The metabolite detection method was adopted from [61]. Briefly, metabolites were separated by Thermo Fisher Vanquish UHPLC with a Waters Acquity UPLC BEH C18 column (1.7 μm, 2.1 × 100 mm; Waters Corp., Milford, MA, USA) and analyzed using a Thermo Fisher Q Exactive orbitrap mass spectrometer in negative ionization mode. LC separation was performed over a 25 min method with a 14.5 min linear gradient of mobile phase (buffer A, 97% water with 3% methanol, 10 mM tributylamine, and acetic acid-adjusted pH of 8.3) and organic phase (buffer B, 100% methanol) (0 min, 5% B; 2.5 min, 5% B; 17 min, 95%B; 19.5 min, 5% B; 20 min, 5% B; 25 min, 5% B, flow rate 0.2 mL/min). A quantity of 12 µL of each sample was injected into the system for analysis. The ESI settings were 30/10/1 for sheath/aux/sweep gas flow rates, 2.50 kV for spray voltage, 50 for S-lens RF level, 350 °C for capillary temperature, and 300 °C for auxiliary gas heater temperature. MS1 scans were operated at resolution = 70,000, scan range = 85–1250 *m*/*z*, automatic gain control target = 1 × 10^6^, and 100 ms maximum IT. Metabolites were identified and quantified using El-MAVEN (v0.12.1-beta) [62] with metabolite retention times empirically determined in-house. Metabolite levels were compared using the peak AreaTop metric.

### 4.3. Statistical Analysis and Bioinformatics

Peak intensity was normalized by median, log transformed, and auto scaled (mean-centered and divided by the standard deviation of each variable) for statistical multivariate analysis. Significantly altered metabolites were identified with a fold change > 1.2 and a *p*-value < 0.05 [24]. These metabolites were presented as volcano plots using GraphPad Prism 9 (GraphPad Prism, La Jolla, CA, USA). Normalized data were evaluated using principal component analysis (PCA), hierarchical clustering heatmap, and KEGG pathway enrichment (mouse library) using MetaboAnalyst 5.0 [24]. KEGG pathways were presented using R version 3.6.3 (www.r-project.org) (accessed on 25 October 2021) with the addition of the ggplot2 package (ggplot2.tidyverse.org) (accessed on 25 October 2021). 

## 5. Conclusions

In this study, we identified and analyzed metabolites in mouse and NMR brain and liver before and after acute hypoxia. We found similar suppression of amino acid-related anabolism in both NMR brain and liver, remodelling of glucose metabolism but with different strategies between NMR brain and liver, an oxidative state shift in NMR brain, and potential evidence for AMPK-regulated metabolic changes in NMR liver. These results are in line with previous reports in NMRs, which demonstrate reduced energy demand to lower O_2_ consumption in hypoxia, and tissue-specific metabolic reorganization. However, our data build upon these previous studies and provide more details of metabolic remodelling that likely support the hypoxia-tolerance of NMRs. Our results reveal potential mechanisms explaining how NMRs reprogram metabolic pathways during hypoxia and support numerous avenues of future exploration in this area of research.

## Figures and Tables

**Figure 1 metabolites-12-00056-f001:**
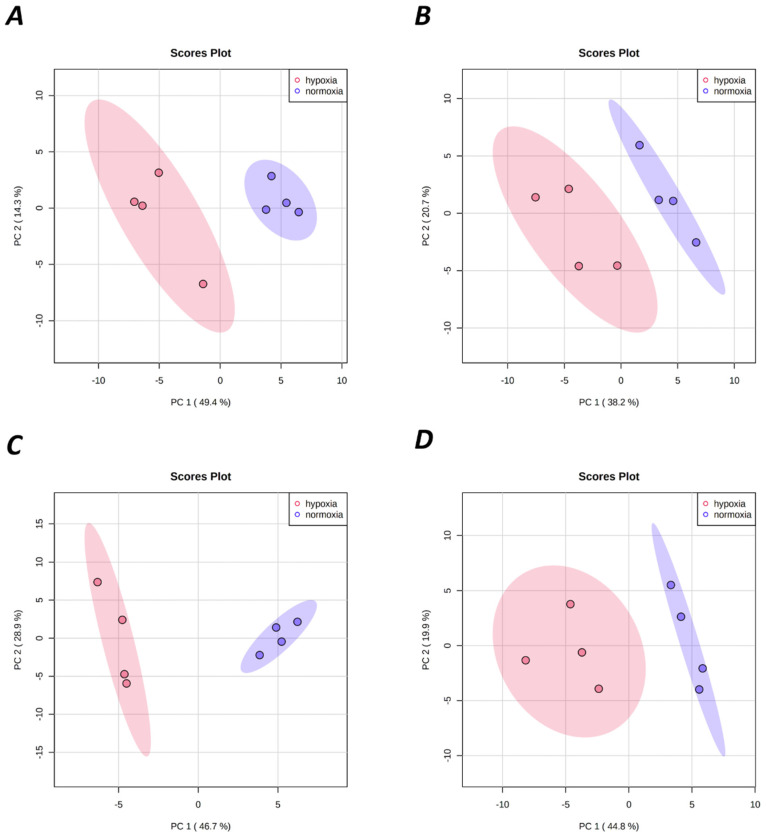
Metabolic profiles in NMR and mouse brain (**A**,**B**, respectively) and liver (**C**,**D**, respectively) during normoxic and hypoxic conditions were visualized through principal component analysis (PCA) between selected PCs. Points represent scores of biological replicates.

**Figure 2 metabolites-12-00056-f002:**
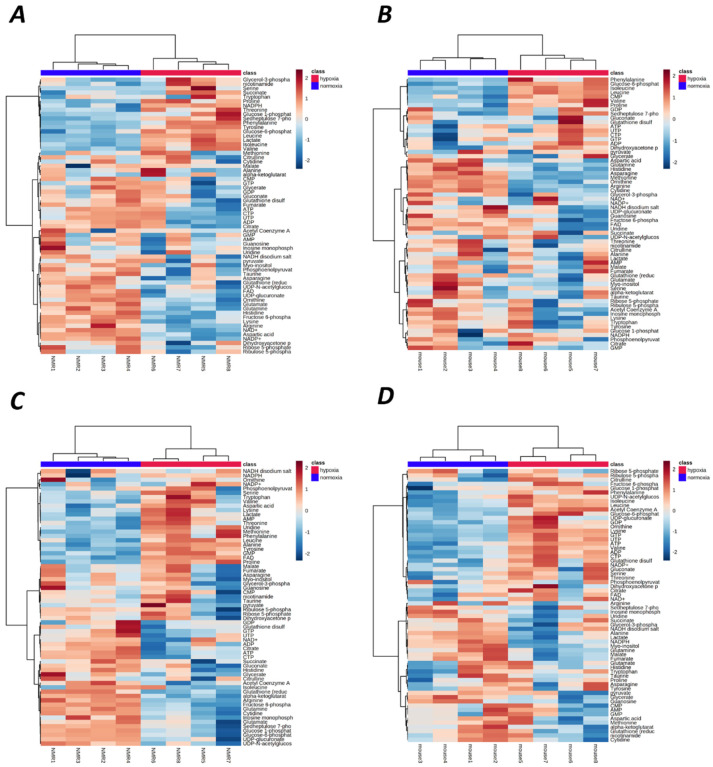
Heat map of metabolite abundance in NMR and mouse brain (**A**,**B**, respectively) and liver (**C**,**D**, respectively). Rows represent individual metabolites, and columns represent biological replicates. Distance was measured using Pearson Correlations and the Ward.D clustering algorithm.

**Figure 3 metabolites-12-00056-f003:**
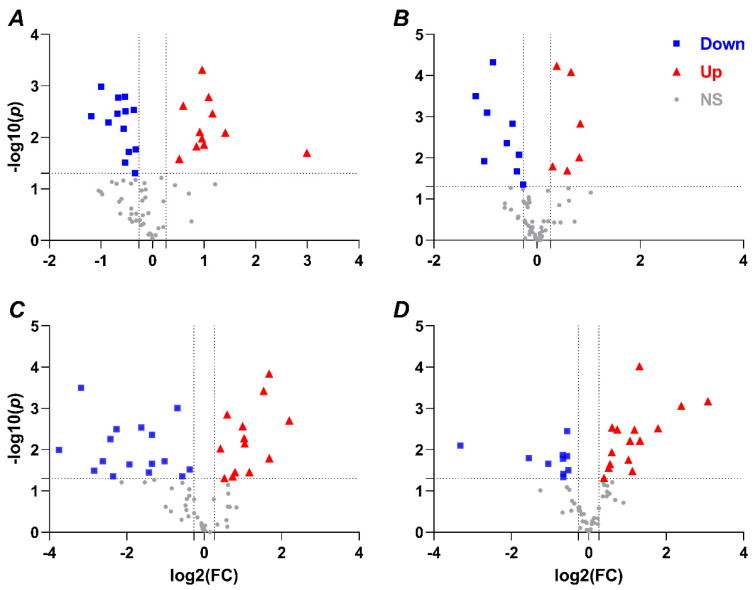
Volcano Plot of the identified metabolites illustrating a comparison between normoxia and hypoxia. Fold change (FC) shown on x-axis: log2(FC); *p* value shown on y-axis: −log10(*p*). Downregulated (Down), upregulated (Up), and nonsignificant changes (NS) are presented with blue squares, red triangles, and grey dots, respectively, from NMR and mouse brain (**A**,**B**, respectively) and liver (**C**,**D**, respectively).

**Figure 4 metabolites-12-00056-f004:**
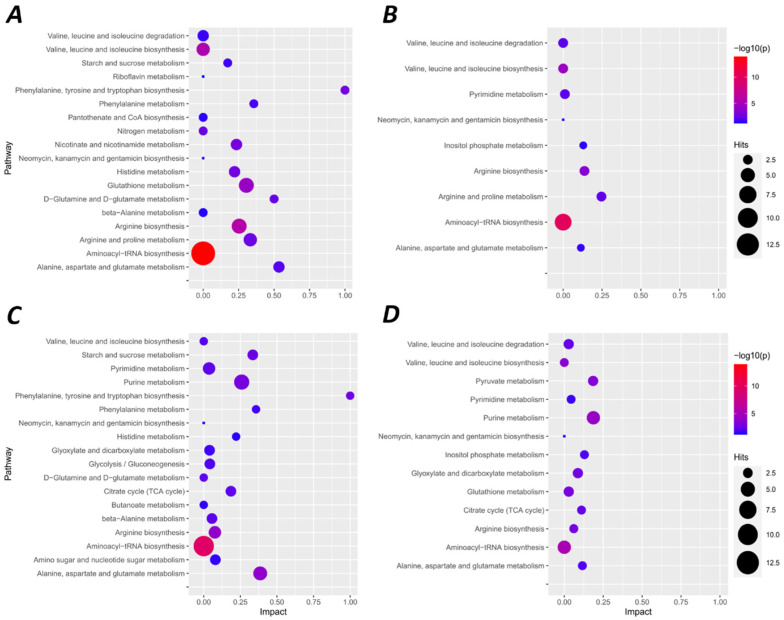
Metabolic pathways in which metabolites are altered by hypoxic treatment shown based on the *p*-value (<0.05), impact factor, and Hits (number of metabolites in pathway) in NMR and mouse brain (**A**,**B**, respectively) and liver (**C**,**D**, respectively).

## Data Availability

Data available in a publicly accessible repository. The data presented in this study are openly available in Figshare at: https://doi.org/10.6084/m9.figshare.16814359 (accessed on 14 October 2021).

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
