# Peer review of "Metabolomic Analysis of Carbohydrate and Amino Acid Changes Induced by Hypoxia in Naked Mole-Rat Brain and Liver"

_metabolites, 2022, doi:10.3390/metabo12010056_

Round 1

Reviewer 1 Report

In this manuscript Hang Cheng et al. compared the fluctuations in the levels of different metabolites in the liver and brain tissue of hypoxia-tolerant naked mole rats (NMR) and mice exposed to hypoxia. They identify several metabolites that are similarly affected in both species while others are altered differentially. Data are interested but the following comments should be addressed

Major comments:

1.- Authors should explain the rationale to use 7% O2 as hypoxic conditions. Authors should consider whether animal movement as well as food intake is reduced when animals (mice or NMR) are exposed to this severe hypoxic conditions. Alterations in movement and food intake – during 4 hours of hypoxic exposure - might potentially explain some of the metabolic data obtained and therefore not primarily attributed to hypoxic stimulus. Authors might perform these experiments with fasted animals just for the 4h period of the hypoxia exposure. Alternatively authors should consider using 10% O2 hypoxic conditions that are better tolerated - at least by mice - and therefore are routinely used to assess mouse response to hypoxia.

2.- In figure 2 authors should compare the same metabolites in brain tissue (A and B) and liver tissue (C and D) to appreciate more clearly, which metabolites are equally affected in NMR and mice and which are differentially altered. Summary table including those metabolites equally affected between species and those differently affected would help to appreciate more clearly the results of the manuscript.  

3.- Similar to the comment 2, authors should compare the same metabolic pathways in brain tissue (A and B) and liver tissue (C and D) to appreciate better the difference between the two species.  

4.- Authors should clarify whether the term ‘post-hypoxia’ refers simply to animals exposed to hypoxia or first exposed to hypoxia and the returned to normoxia.

Author Response

Replies to the first reviewer

We would like to take this opportunity to thank you for your kind and helpful comments that have helped us to substantially improve our submission. All comments and suggestions have been addressed as detailed below:

Reviewer #1 overall comments: In this manuscript Hang Cheng et al. compared the fluctuations in the levels of different metabolites in the liver and brain tissue of hypoxia-tolerant naked mole rats (NMR) and mice exposed to hypoxia. They identify several metabolites that are similarly affected in both species while others are altered differentially. Data are interested but the following comments should be addressed.

Major comments:

  1. Authors should explain the rationale to use 7% O2 as hypoxic conditions. Authors should consider whether animal movement as well as food intake is reduced when animals (mice or NMR) are exposed to this severe hypoxic conditions. Alterations in movement and food intake – during 4 hours of hypoxic exposure - might potentially explain some of the metabolic data obtained and therefore not primarily attributed to hypoxic stimulus. Authors might perform these experiments with fasted animals just for the 4h period of the hypoxia exposure. Alternatively authors should consider using 10% O2 hypoxic conditions that are better tolerated - at least by mice - and therefore are routinely used to assess mouse response to hypoxia.

Response: We chose 7% O2 as our hypoxic treatment condition because it provides a strong hypoxic challenge in both study species. In our previous studies we have found that NMRs have relatively minor metabolic, physiological, and behavioural responses to >9% O2 and we wanted to ensure that both species were robustly challenged by our hypoxic treatment. Furthermore, most previous studies exploring mechanisms of hypoxia-tolerance in NMRs use 7% O2 (or lower) and we are primarily interested in this species, thus we chose to explore the level of O2 in which NMRs are most often assessed, instead of that in which mice are often assessed, which was of lesser interest to our aims.

      We agree that behaviour and diet are certainly impacted by hypoxia in both species, and further that this impact would occur at most levels of hypoxia exposure, including 10% O2. However, these are still impacts of the hypoxic exposure. Indeed, we expect a very wide range of cellular and physiological changes in hypoxia, from alterations of protein synthesis and DNA transcription and urea cycling, to changes in food intake and release of glucose into the blood from hepatic stores, to changes in behaviour, heart rate, blood O2 carrying capacity, etc. These changes have enormous impacts on tissue-level metabolic demand, circulation of nutrients and O2 in the blood, etc. Therefore, the hypoxic stimulus is responsible for ALL the metabolic changes we observe, although certainly through myriad pathways and mechanisms. It is thus not possible (or meaningful) to assess the impact of hypoxia on metabolism in isolation from the impact of hypoxic-mediated changes in physiological function on metabolism; they are one and the same.

  1. In figure 2 authors should compare the same metabolites in brain tissue (A and B) and liver tissue (C and D) to appreciate more clearly, which metabolites are equally affected in NMR and mice and which are differentially altered. Summary table including those metabolites equally affected between species and those differently affected would help to appreciate more clearly the results of the manuscript.

Response: We are primarily interested in the between-species comparison and in what metabolites differ; however, we have revised our manuscript to moderately increase our focus on common patterns of change in NMR brain and liver, and in NMR and mouse brain.

  1. Similar to the comment 2, authors should compare the same metabolic pathways in brain tissue (A and B) and liver tissue (C and D) to appreciate better the difference between the two species.

Response: Please see our response to comment 2, above.    

  1. Authors should clarify whether the term ‘post-hypoxia’ refers simply to animals exposed to hypoxia or first exposed to hypoxia and the returned to normoxia

Response: We have revised our manuscript to remove this term and instead refer to “during hypoxia” or “in hypoxia” to more clearly indicate that samples were taken from animals that were treated in hypoxia alone, and not allowed to recover in normoxia after treatment.

Reviewer 2 Report

The authors submitted an interesting and well-written paper.

In Figure 3, the Volcano Plot of the identified metabolites comparison between normoxia and hypoxia are presented. In the plots, the authors used different colors of symbols. I suggest also using different symbol shapes, which in the greyscale will make identifying groups easier.

Report on Metabolic analysis of carbohydrate and amino acid changes induced by hypoxia in naked
mole-rat brain and liver by Cheng, H. et al.
In the manuscript the authors report on metabolite changes caused by hypoxia in the brain and liver
tissue of naked mole-rat (NMRs) and mice. NMRs are highly tolerant to hypoxia. The authors
compare metabolic pathways and try to identify factors relevant for the different post-hypoxia
hallmarks of studied tissues.
Major comments:
The authors do not explain why they used 7% of oxygen concentration to create hypoxic conditions.
The authors claim that NMRs do not exhibit circadian rhythmicity of general locomotor activity but
they do not comment on possible mice movement and it’s impact on their metabolism.
In figure 4, the altered metabolites post hypoxic treatments are shown. I would suggest to unify
both log scales of the p-value and the impact scales. The hits legend seems to be nonuniform as
well. Additionally, the authors limited the content of Fig B and D only to the metabolites altered in
the species. This approach makes it more difficult to the reader to compare rat to mice data.

Author Response

Replies to the second reviewer

We would like to take this opportunity to thank you for your kind and helpful comment that has helped us to improve the figures in our submission.

Reviewer #2 overall comments: In the manuscript the authors report on metabolite changes caused by hypoxia in the brain and liver tissue of naked mole-rat (NMRs) and mice. NMRs are highly tolerant to hypoxia. The authors compare metabolic pathways and try to identify factors relevant for the different post-hypoxia hallmarks of studied tissues. The authors submitted an interesting and well-written paper.

Major comments:

  1. The authors do not explain why they used 7% of oxygen concentration to create hypoxic conditions.

Response: Please see our response to Reviewer 1, question #1,

  1. The authors claim that NMRs do not exhibit circadian rhythmicity of general locomotor activity but they do not comment on possible mice movement and it’s impact on their metabolism.

Response: To be clear, we state (not claim) that NMRs do not exhibit circadian rhythmicity of general locomotor activity based on the evidence provided by previous studies from other laboratories. We expect that this is not the case in mice; however, we did conduct all experiments and sacrifice all animals at the same time of day to limit the impact of circadian rhythmicity on our data. In other words, both the control and treatment animals were treated at the same time and circadian rhythmicity should therefore have the same impact on both groups. We have modified our methods section to point out that animal sacrifices were conducted consistently in both species. We also expect that hypoxia will impact behaviour and activity in both species, likely reducing activity in general. However, this is part of the impact of hypoxia and we wanted to measure metabolic changes that occur alongside previously documented behavioural and physiological changes in NMRs. Please also see our response to Reviewer 1, question 1, above.

  1. In figure 4, the altered metabolites post hypoxic treatments are shown. I would suggest to unify both log scales of the p-value and the impact scales. The hits legend seems to be nonuniform as well. Additionally, the authors limited the content of Fig B and D only to the metabolites altered in the species. This approach makes it more difficult to the reader to compare rat to mice data.

Response: Thank you for your comment. In response, we now present Fig. 4 with common scales for impact, p-value, and dots sizes of Hits. To make our figures and the text less cumbersome, we utilized a common cut-off (p<0.05) for all of four tissues and we chose to highlight metabolites pathways that were altered in Fig. 4. However, please note that all data summaries are included in the supplementary file and all raw data is available via FigShare. Thus, the interested reader is able to view the full list with ease, or to delve into the raw data, if they so wish.

Minor comment:

  1. In Figure 3, the Volcano Plot of the identified metabolites comparison between normoxia and hypoxia are presented. In the plots, the authors used different colors of symbols. I suggest also using different symbol shapes, which in the greyscale will make identifying groups easier.

Response: We have revised this figure as suggested, i.e., using blue squares for decreases, red triangles for increases and grey dot for data with no changes.

Reviewer 3 Report

In the present work, Cheng and colleagues compare the carbohydrate and amino acid changes induced by hypoxia in naked mole rats and mice. Their study is well conducted and all the experimental results are explained in light of the literature. Although it can not be considered a groundbreaking job it is well performed and, in my opinion, no changes are needed before its publication. 

Author Response

Replies to the third reviewer

Reviewer #3 overall comments: In the present work, Cheng and colleagues compare the carbohydrate and amino acid changes induced by hypoxia in naked mole rats and mice. Their study is well conducted and all the experimental results are explained in light of the literature. Although it can not be considered a groundbreaking job it is well performed and, in my opinion, no changes are needed before its publication.

Response: Thank you for your efforts in peer-reviewing our manuscript!

Round 2

Reviewer 1 Report

Authors have responded satisfactorily to my comments. However the following additional comments should considered.

1.- Regarding my first previous comment:

  • Authors might explain the rationale to use 7% O2 as hypoxic conditions. For example authors might include references showing that NMR are really exposed in nature to these severe hypoxic conditions.

  • Authors mentioned “We agree that behaviour and diet are certainly impacted by hypoxia in both species, and further that this impact would occur at most levels of hypoxia exposure, including 10% O2”. However this is not entirely correct since mice exposure to 7% O2 have a more profound impact than 10%O2

  • Authors might state in the manuscript (in the discussion section for example) that metabolic changes observed could be a secondary consequence of reduced food intake and not to a primary response to the hypoxia exposure.

Author Response

Dear Reviewer, thank you for your further comments. Please note, however, that animals did not receive food during the treatment conditions (either in normoxia or hypoxia) and so there was no food intake during these experiments in either treatment group. Our analysis evaluates differences in metabolite expression between these treatment groups and since food intake has been removed as a confounding variable due to our experimental design, we do not believe that this factor is relevant or appropriate to discuss in the body of our manuscript. 

Round 3

Reviewer 1 Report

The authors have addressed satisfactorily my comments. However authors should state in the manuscript (methods section) that animals were fasted during the normoxic or hypoxic treatment conditions.

Author Response

We have included a comment about the animals not being provided with food during experiments in the methods section.